# Distributed-Framework Basin Modeling System: I. Overview and Model Coupling

Chuanhai Wang [1,2], Wenjuan Hua [1,2], Gang Chen [1,2,*], Xing Fang [3] and Xiaoning Li [1,2]

1    State Key Laboratory of Hydrology-Water Resources and Hydraulic Engineering, Hohai University, Nanjing 210098, China; chwang@hhu.edu.cn (C.W.); huawenjuan0106@126.com (W.H.); xzl0938@hhu.edu.cn (X.L.)
2    College of Hydrology and Water Resources, Hohai University, Nanjing 210098, China
3    Department of Civil and Environmental Engineering, Auburn University, Auburn, AL 36849-5337, USA; xing.fang@auburn.edu
*    Correspondence: gangchen@hhu.edu.cn; Tel.: +86-139-1302-9378

**Abstract:** To better simulate the river basin hydrological cycle and to solve practical engineering application issues, this paper describes the distributed-framework basin modeling system (DFBMS), which concatenate a professional hydrological model system, a geographical integrated system, and a database management system. DFBMS has two cores, which are the distributed-frame professional modeling system (DF-PMS) and the double-object sharing structure (DOSS). An area/region that has the same mechanism of runoff generation and/or movement is defined as one type of hydrological feature unit (HFU). DF-PMS adopts different kinds of HFUs to simulate the whole watershed hydrological cycle. The HFUs concept is the most important component of DF-PMS, enabling the model to simulate the hydrological process with empirical equations or physical-based submodules. Based on the underlying source code, the sharing uniform data structure, named DOSS, is proposed to accomplish the integration of a hydrological model and geographical information system (GIS), which is a new way of exploring temporal GIS. DFBMS has different numerical schemes including conceptual and distributed models. The feasibility and practicability of DFBMS are proven through its application in different study areas.

**Keywords:** distributed-framework; double-object sharing structure; hydrological model; river basin modeling

## 1. Introduction

### 1.1. The Issue of the Distributed Hydrological Model

Watershed hydrological modeling is an important approach for simulating and understanding watershed hydrologic processes [1]. From the initial blueprint proposed by Freeze and Harlan in 1969 (FH69) [2], distributed hydrological models have been developed for more than 50 years. The distributed basin hydrological model can better represent the impact of soils, vegetation cover and land-use for the runoff process, which has become the development direction of the hydrological model [3,4]. In the last few years, with the development of Information Communication Technologies (ICTs) [5], such as computer science, remote sensing, and geographical information system (GIS), the cost of basin spatial information data, such as terrain, soil, and vegetation type data, has become lower and lower. An increasing number of distributed models are using physical mechanism-based equations to describe rainfall, snowmelt, evaporation, interception, infiltration, soil water movement and other physical processes. These models, such as the Systeme Hydrologique Europeen (SHE), soil and water assessment tool (SWAT), and distributed Xinanjiang model, have already been widely applied and validated [6–9]. The SHE model is regarded as the first distributed hydrological model and was jointly developed by the Danish Hydraulic Institute, British Institute of Hydrology, and Sogreah. The SHE model adopts the partial

differential equations of mass conservation, momentum conservation, and energy conservation (such as Rutter, Penman–Monteith, St. Venant, Richards, Boussinesq, etc.) to simulate the rainfall–runoff process, runoff concentration in the channel, saturated and unsaturated soil water movement and other hydrologic cycles. Meanwhile, a distributed conceptual model describes the watershed water cycle in different levels and has also been widely used for its simplified structure and parameter. It can be easily to calibrate and meet the requirement of the real-time forecast, such as the distributed Xinanjiang model [9] and SWAT model [10]. Similarly, other spatial distributed models that simulate the hydrological cycle through dedicated inherent modules and that have been applied in various spatial and temporal scales are also described in the literature [11,12].

Compared to the lumped model, the distributed model has obvious advantages in model structure and physical meaning of parameters, but it cannot always yield better results [13–15]. In the Distributed Model Intercomparison Project (DMIP), researchers aimed to understand how to more reasonably use high-precision remote sensing information and GIS data in flood forecasting, or under what conditions distributed model can provide more accurate simulations. The parameter estimation problem is a bigger challenge for distributed hydrologic modeling than lumped hydrologic modeling. The simulation results from twelve distributed models (the SWAT, SAC-SMA, MIKE11, NOAH Land Surface, HRCDHM, Tribs, HL-RMS, r.water.fea, VIC-3L, TOPNET, WATERLOO, and LL-II models) were compared with observed data and a lumped model (SAC-SMA) [13,16]. The lumped model provided better simulation results than the distributed models [13,15,16]. Beven [17] pointed out that the main problem of a distributed hydrological model based on the FH69 blueprint is scale issues. The equations describing the hydrological processes are usually based on mass and energy conservation on a point scale. However, the models are always applied on a large-scale grid, such as grids of 50 m, 100 m, or even larger [18]. Meanwhile, the models usually use uniform assumptions to describe the change of internal heterogeneity in the computing unit, which leads to different simulations being conducted at different spatial and temporal scales. Kavvas and Levent [19] attempted to develop some general conservation equations for the probability distributions and means (ensemble averages) of hydrologic processes that are governed by nonlinear partial differential equations such as point location scale.

As always, hydrologists aim to understand the temporal and spatial distribution characteristics of hydrological models, although many issues still exist. On the one hand, the current lumped models and distributed conceptual models simulate the watershed water cycle conceptually and with generalization, especially for the rainfall-runoff process. The problem of basin heterogeneity has not been well considered, which leads to poor performance for spatial distribution generalization. On the other hand, although the distributed physical model has made great progress, most models have complex structures. Additionally, there are lot of parameters with unclear physical meanings that need to be calibrated. Therefore, the distributed physical model is difficult to apply in non-data areas. Current problems associated with the distributed physical model, distributed conceptual model, and lumped model depend on the improvement of perceiving the hydrological process and related fields. Given the current cognitive level and science conditions, combining and taking advantage of different kinds of models represent an important research aspect for the basin hydrological model. To better understand the impact of the vegetation cover and soil dynamics on the hydrological process, the hydrological models are usually integrated with the geographic information system (GIS).

### 1.2. The Issue of the Integration of the Geographic Information System and Hydrological Models

In the 1960s and 1970s, the technologies employed for the geographic information system GIS and professional models of water were developed independently [20]. In the late 1980s, researchers started to work on the integration of GIS and hydrological models to meet the requirement of GIS function analysis [21]. On the other hand, more and more projects and studies began to need precise geographic information [22]. Goodchild [23]

thought that the integration of GIS and the hydrological model could be an important part of improving geospatial analysis and modeling capabilities. Payne [24] used absolute space generalization to set space in the geometry index and time in a discrete timeslice. The study met many requirements, but it broke the continuity of geographic objects, which could lead to missing geophysical events in the sequence. Therefore, representation and modeling that support complex geographical and continuous objects have been hot research topics [25].

GIS is widely used to build systems of database management and decision support at different spatial scales [26]. The development of distributed hydrological models is becoming increasingly dependent on GIS. There are four different ways to integrate hydrological models and GIS [21]: (a) embedding GIS in the hydrological model, such as RiverCAD, HEC-RAS (version 5.0.4 and later), RiverTools, and MODFLOW; (b) embedding the hydrological model in GIS, such as ArcGIRD and Arc Hydro from ESRI, Redlands, CA, USA; (c) a loosely coupled model that is integrated using independent software; and (d) a tightly coupled model with GIS and a hydrological model with a customized unified interface achieved by combining functions of different software. However, all these integrations are only technology-driven, that is to say, the integrations result from coupling based on the data form, not the internal structure. This represents low-level coupling, which has the following problems:

(a) The issue of spatial-temporal characteristics. Hydrologic elements vary over time, that is to say, the hydrological process exhibits procedural change with time. The characteristic of temporal and spatial change is an essential basis for hydrological analyses and simulation. However, traditional GIS focuses on expressing and analyzing spatial data and attribute data that lack temporal aspects for space. Temporal GIS has been proposed for a few years, but is still in the stages of theoretical and model studies. GIS is commonly used to solve the topographical model, which does not change with time. This represents a serious impediment to integrating hydrological models and GIS;

(b) The issue of topological relations. Topology is an important basis for research on how to associate geographical entities in space. In GIS, geographic data are composed of positioning feature, attribute feature, and topological feature data. The positioning feature and topological data are the spatial features, which record the spatial structural relationship between objects. Hydrological data models are based on the topological relation of a node-arc-polygon. Traditional GIS has good applicability based on the common expression of geographical information. Hydrology belongs to a relatively professional field that usually studies complex geographic objects, such as general topographic regions and channel topographic regions. Complex geographic objects are composed of points, lines, and surfaces from simple GIS objects. They have a specific topological relationship inside, such as the location between the upstream and downstream cross-sections for a channel. In terms of the traditional topological relations of GIS [27], it is very complicated to describe complex objects of hydrology and hard to satisfy the complex hydrological analyses. Therefore, it is very important to solve the topological relations of complex objects for the hydrologic model;

(c) The issue of deficiency for analysis. The model of terrain statistical analysis is well-developed in GIS, which contains a digital elevation model, a spatial statistics and analysis model, a path analysis model, an overlay analysis model, and so on. However, all of the models are nonprocedural static models that show a deficiency in the process dynamic model. The watershed resource management models are mainly dynamic. This represents the issue of temporal GIS—it is hard to extend the analysis function of GIS for exsiting distributed hydrological models since they do not solve the application of procedural dynamic models.

This series of four papers aims to introduce/develop the modeling system framework/structure, theories and methods of hydrological/hydraulic modeling, and various application case studies of the distributed-framework basin modeling system (DFBMS). The whole series contains four parts: (I) overview and model coupling; (II) hydrologic modeling system; (III) hydraulic modeling system; and (IV) application in Taihu Basin.

This first paper focuses on the overview and system integration for DFBMS. The structure of DFBMS consists of a professional model system (i.e., hydrologic and hydraulic modeling system in the study), double-structure GIS, and database management system. The professional model system is the core part of DFBMS, which contains a distributed-frame hydrological modeling system (DF-HMS) and distributed-framework river modeling system (DF-RMS) that adopt different submodules to simulate the whole watershed hydrological cycle. The double-structure GIS is proposed to solve the sharing issues between the complex geographical object and professional model object, which is a new way of exploring temporal GIS. The DFBMS has great advantages and efficiencies in modeling hydrologic and hydraulic responses in non-homogenous catchments.

## 2. Materials and Methods

### 2.1. The Structure of the Distributed-Framework Basin Modeling System

In hydrology research, the digital basin model is widely used, and is the application subsystem employed for the digital earth. It has two important cornerstones at the level of the digital earth [28]. One is the information highway and high-speed wideband network technology, and the other one is spatial information technology and infrastructure. The application of the digital basin model requires lots of information to be collected and transmitted, which is due to the construction of infrastructure projects. The digital basin model should be a software system focusing on the fusion of professional models, visualization presentation, and information management. Within the designed system framework, it is possible to access spatial information through high-speed wideband network. At this stage, the data need to be imported manually, such as from AutoCAD (from AUTODESK, San Rafael, CA, USA), ArcGIS (from ESRI, Redlands, CA, USA), MapInfo (from Syncsort, North Greenbush, NY, USA). In this study, the structure of the distributed-framework basin modeling system (DFBMS) is proposed as one kind of digital basin model. The structure of DFBMS is shown in Figure 1.

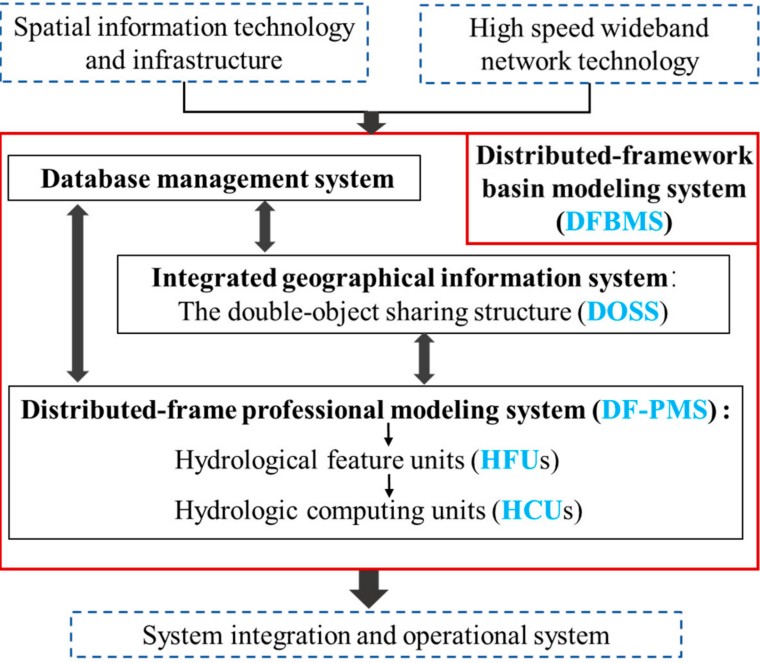

**Figure 1.** The structure of the distributed-framework basin modeling system (DFBMS).

The design of DFBMS is based on information collection and transmission, which carries out the applications of information and visualization. The National Spatial Information Infrastructure provides the basic data for DFBMS, such as terrain, digital elevation, and geographic feature data from resource satellites and remote sensing (RS). High-speed

wideband network technology is another infrastructure construction supporting data transmission for DFBMS. DFBMS consists of three parts, shown in the red rectangular box of Figure 1:

a   Database management system: The basic system for DFBMS. This contains the technology for data management, mass data memory and management, data mining, and the data warehouse.

b   Geographical integrated system: The core part of DFBMS that carries out the visualization of model results and information. The capabilities and functions of GIS in the DFBMS are completely self-developed, which is the main role of the geographical integrated system. It not only provides the spatial information required by DFBMS, but also supplies a visual representation of spatial information and model results. The GIS can interpret remote sensing data as being one of four different types of underlying surface that will be used for rainfall runoff.

c   Professional model system: A modeling system that can be used to simulate changes in the geographical environment in the past, present, and future. The professional model system is another kernel for DFBMS, containing various professional models, such as the hydrologic model, hydraulic model, digital basin generating model, water quality model, and sediment yield model. In this study, we only introduce the hydrologic and hydraulic model.

DFBMS is not only designed for theoretical research, but also for solving practical problems. For example, performance in real-time is an important requirement in flood forecasting. When integrated with DFBMS, the operating system can be used to solve the practical issues in support of model application and presentation. For example, through database management system, the DFBMS can access rainfall and evaporation forecast data from different institutions. The simulation and forecast of water surface elevation, discharge are uploaded to the database. Eventually, it becomes an industry system platform in a specific area, such as digital water and digital flood control systems.

## 2.2. Distributed-Framework Professional Modeling System (DF-PMS)

### 2.2.1. Hydrological Processes

The process of the basin water cycle can be separated into a vertical cycle and longitudinal cycle (Figure 2). The basic principles of the water cycle in different stages are diverse in both directions. In the vertical cycle, water is transferred and transformed via different mediums in different states (Figure 2a). There are two main stages in the vertical cycle, one of which is the phase of water vapor migration and transformation in the air, which belongs to the research field of meteorologists. Water vapor enters the atmosphere through evapotranspiration from the ocean, inland water bodies, plants, soils, and construction land, and then returns to the inland area through atmospheric migration and transport, before falling on different areas. Precipitation includes rainfall and snowfall, which are the input data in this study. The other stage involves the circulation of water on land, which is relatively sufficiently studied by hydrologists, such as circulation from the land surface to unsaturated soil water with a high and not well known heterogeneity.

In the longitudinal cycle, the movement and transformation of water are mainly affected by the land surface topography and cover, characteristics of the underlying surface, and underlying soils. It can be decomposed into two stages: Runoff generation and flow movement (i.e., confluence on the land surface and routing in rivers/channels or underground pipe networks). Runoff generation mainly depends on land surface conditions (i.e., soil surface humidity, soil surface vegetation cover, soil surface compaction, etc.), irrespective of whether they are under hilly, plain, or tidal areas. The movement of water flows from high to low land, i.e., from hilly areas (sub-watersheds or rivers) to plain areas, and then arrives in tidal areas (Figure 2b). The mechanism of the runoff is dfifferent between uptream and donwstream areas. In hilly areas, the confluence time is short due to the steep terrain. The flow of the river system typically occurs in one direction with a dendritic shape by stream orders [29], and has a unique outlet section. There are no or few interactions

between the mainstream and tributaries. However, in plain areas, the terrain exhibits little variation. The mainstream and tributaries interact with each other in a crisscrossed and network-like manner. The flow direction is uncertain and depends on various factors, especially for hydraulic engineering. Lakes, flood plains, and paddy fields are also scattered over plain areas where the flooding water can flow into or be discharged. Therefore, flood spread and propagation are complex in plain areas. The downstream section of the longitudinal cycle is the tidal area (estuaries, wetlands, and marshlands), and the flood movement is even more complex due to the action of tidal and seawater jacking, which can be coupled with the control of manmade structures and buildings. Overall, the laws of runoff generation and movement of different areas in the longitudinal direction are quite different, and these make it impossible to describe runoff/flow movement with a simple unanimous theory. Due to the wide coverage, complex underlying surface, numerous influencing factors, and changeable medium, the laws of the land surface water cycle have not yet been fully understood.

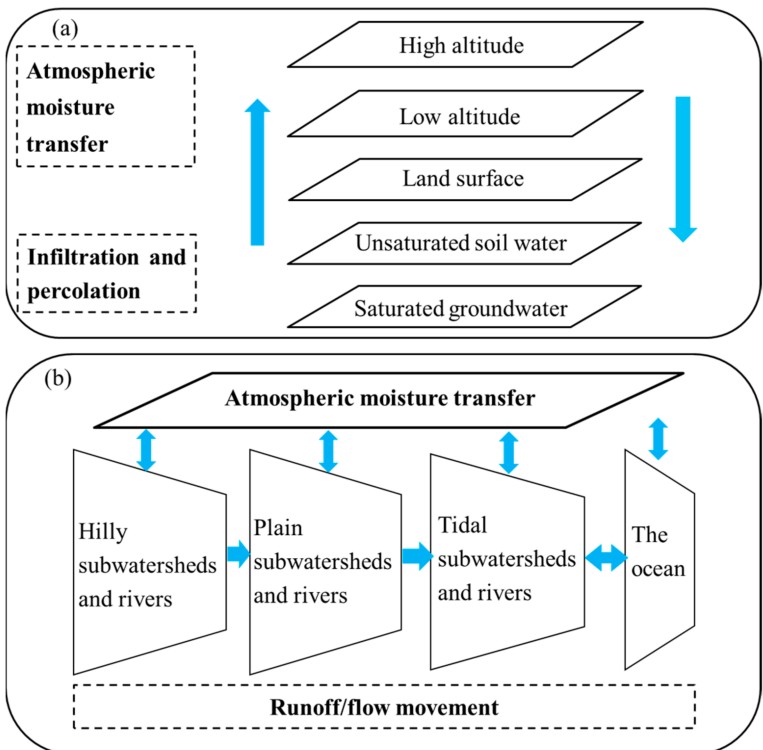

**Figure 2.** Schematic diagram of the water cycle in (**a**) vertical and (**b**) longitudinal directions.

### 2.2.2. Distributed-Frame Professional Modeling System

The distributed-frame professional modeling system (DF-PMS) is designed to simulate all hydrological processes, such as snowmelt runoff, slope runoff, river network flow, soil unsaturated flow, saturated flow, and surface and groundwater exchange (Figure 3). The primary hydrological process can be described as rainfall-runoff generation and transfer under different underlying surfaces. Hydrological feature unit (HFU) is defined as an area/region that has the same mechanism of runoff generation and/or movement. According to the concept of HFU (Table 1), the research area can be divided into corresponding HFUs and coupled to describe the whole watershed hydrological cycle, which can deal with non-homogeneous catchments/basins. DF-PMS includes two modeling systems: The distributed-framework hydrologic modeling system (DF-HMS), which will be described in the second paper in this series, and the distributed-framework river modeling system (DF-RMS), which will be described in the third paper in this series. DF-PMS is not always a full-scale distributed hydrologic model system, and can be formed of conceptual models or black box models for different HFUs, as required, which is different from the concept of the

traditional distributed hydrologic model. For example, the concept model can be applied if it is a non-data area. However, if there are a lot of observations for the study area, then the distributed physical model can be used to simulate the hydrological process. The HFU concept is the most important component for DF-PMS, enabling the model to simulate the hydrological process through conceptual formula or physical-based submodules, as required. At present, DF-PMS mainly focuses on the water cycle simulation on the land surface and soil. For observed or simulated rainfall and snow, they are set as the input data. As mentioned before, HFUs can be further divided into hydrologic computing units (HCUs) to reflect the inhomogeneity of underlying spatial components.

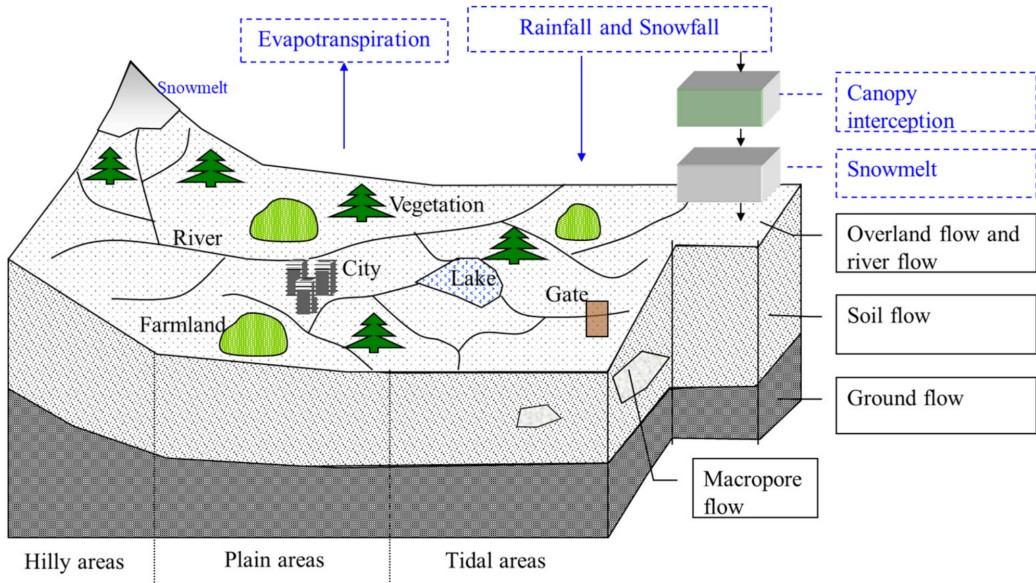

**Figure 3.** A schematic diagram for distributed-framework hydrological processes in the water cycle.

The specific DF-PMS for the study area can be built when the HFUs are fixed and corresponding solving models are selected. The coupling of different HFUs represents multi-basin, multi-scale, and multi-process fusion in the hilly, plain, and tidal area, which covers not only hydrological processes, but also hydrodynamics. It is necessary to consider the time scale for the internal boundary conditions of each HFU. The major concern for the coupling of HFUs is the water exchange between the interface of different HFUs. When the water exchange occurs in one direction during the period or the variation of the time scale is large, the explicit coupling mode is applied for some interfaces of HFUs, such as the runoff generation of hilly sub-watershed HFU enter into the nearby hilly river HFU. In other words, the explicit coupling mode can be used between HFUs of the runoff generation type or between HFUs of the runoff and confluence type. However, for HFUs of runoff movement, the water exchange is frequent and the time scale is small. We have to apply the implicit coupling mode, especially between HFUs in the surface layer, such as the river and lake feature unit and the weir-sluice feature unit in plain areas. For the confluence unit between different vertical stratification, such as groundwater, the explicit coupling mode is applied due to the large time scale and one-way exchange in a certain period.

**Table 1.** Typical hydrological feature units (HFUs) in a basin.

| No. | HFUs | Type | Influencing Factors | Discrete Scale in Simulation | | Vertical Circulation | Transverse Circulation | Simulation Models or Methods | Primary Reference |
|---|---|---|---|---|---|---|---|---|---|
| | | | | Time Scale | Spatial Scale (HCUs Division) | | | | |
| 1 | Snowfield | Runoff generation | Snow depth, area, terrain, perennial average temperature, meteorological condition, etc. | 1 day | Distributed according to the actual type, no more division for the HCUs. | Snowmelt | Hilly areas | The energy-balance method, or the diurnal temperature-index method | [30,31] |
| 2 | Hilly sub-watershed | Runoff generation and confluence | Subwatershed, landuse, collecting area, soil type, vegetation type, etc. | hour, up to day | The area according to the time scale, and the HCU division based on the distribution of underlying information, usually in kilometers. | Overland flow and river flow, soil flow, ground flow | Hilly areas, plain areas | Xinanjiang model, or TOPMODEL | [7,9] |
| 3 | Hilly river | Confluence | Flow regime (sub- or supercritical flow) | hour | The reach length for HCUs (cross-section) according to the time scale, usually the distance for cross sections is in kilometers. | Overland flow and river flow | Hilly areas, plain areas | Muskingum model | [32] |
| 4 | Plain overland flow | Runoff generation and confluence | Subwatershed, landuse, collecting area, soil type, vegetation type, etc. | 1 hour to 1 day | The area according to the time scale, and the HCU division based on the distribution of underlying information, from meters to kilometers. | Overland flow and river flow, soil flow, ground flow | Plain areas, tidal areas | Rainfall-runoff model in water area, rain-fed land, paddy field and construction land | In Section 2.2.3 of the 2rd paper in this series |
| 5 | Plain river | Routing | Flow regime, flow direction, hydraulic engineering structures, intake and drainage, etc. | seconds to hours | The reach length for HCUs (cross-section) according to the time scale, usually the distance for cross-sections is from meters to kilometers. | Overland flow and river flow | Plain areas, tidal areas | One or two dimensional hydrodynamic model | In the 3rd paper in this series |
| 6 | Urban pipe network | Routing | Distribution of pipe network, design parameter, sediment in the pipe, etc. | seconds to hours | Spatial scale according to major inspection well, usually from meters to kilometers | Overland flow and pipe flow | Plain areas, tidal areas | Urban pipe network model | Further paper |
| 7 | Lakes and reservoirs (including flood plains and paddy fields) | Routing | Natural topography, wind speed, etc. | seconds to hours | Usually from the meters to kilometers for two-dimensional simulation, but no grid for zero-dimensional simulation | Overland flow | Hilly areas, plain areas, tidal areas | Zero-, or two dimensional model | In the 3rd paper in this series |
| 8 | Hydraulic engineering structures | Routing | According to type of engineering, overflowing type, etc. | Same as connected hydrological feature unit | Same as the engineering scale | Overland flow and river flow | Hilly areas, plain areas, tidal areas | Gate and dam simulation | In the 3rd paper in this series |
| 9 | Unsaturated soil water zones | Runoff generation | Soil type, initial water content, etc. | minutes to hours | According to different calculated modes, from meters to kilometers | Unsaturated soil water | Hilly areas, plain areas, tidal areas | Finite element subsurface FLOW system (FEFLOW) | [33] |

**Table 1.** *Cont.*

| No. | HFUs | Type | Influencing Factors | Discrete Scale in Simulation | | Vertical Circulation | Transverse Circulation | Simulation Models or Methods | Primary Reference |
|-----|------|------|---------------------|------------|---------------------------|----------------------|------------------------|------------------------------|-------------------|
| | | | | Time Scale | Spatial Scale (HCUs Division) | | | | |
| 10 | Saturated groundwater zones | Confluence | According to the groundwater table: Deep or shallow | 1 day to 1 month | Grid divided according to the distribution of underground rock | Saturated groundwater | Hilly areas, plain areas, tidal areas | FEFLOW | [33] |
| 11 | Karst regions | Confluence | According to macropore flow and underground river region | 1 h | The area according to the time scale, and the HCU division based on the distribution of underground river region | Interflow | Hilly areas | Not applied in this stage | – |

2.2.3. Hydrological Feature Unit (HFU)

The hydrological feature unit (HFU) concept is an aspect of the professional model system that is proposed as a distributed-frame professional modeling system (DF-PMS) in this study. HFU is defined as an area/region that has the same mechanism of runoff generation and/or movement. The runoff movement on a watershed scale or for overland flow is called runoff transfer, and in river systems (rivers, lakes, reservoirs, flood plains, paddy fields, and through hydraulic structures, including underground pipe networks) this is called routing. The HFU has been classified to include four categories: The runoff generation type; confluence type; mixed runoff generation and confluence type; and routing type (Table 1). DF-PMS currently includes 11 kinds of HFUs, as summarized in Table 1. The water movement is different on the surface, in the soil, and underground for spatial variations of runoff generation. In mountainous areas, the flow of the outlet section controls the confluence for the sub-basin, which is regarded as hilly sub-watershed HFU and hilly river HFU. In the plain river network area, there are several outlet sections, rather than one outlet section for the sub-basin, which are regarded as plain overland-flow HFU and plain river HFU. The flood retention area is treated as the lake and reservoir HFU. Additionally, there is an urban pipe network HFU for describing hydrograph routing in cities. The hydraulic engineering structure HFU is used to connect river HFUs or lake HFUs, such as weirs, gates, and culverts. For the confluence in the soil and underground, there are saturated groundwater zone and karst region HFUs.

Different HFUs have different mechanisms for runoff generation and/or movement. However, for one kind of HFU, different computing methods/modules can be used for this HFU. For example, when a hilly sub-watershed HFU is applied in two different hilly areas, one area can be simulated with the lumped model [34], and the other area can be simulated with the distributed Xinanjiang model [35]. The simulation methods are chosen according to the requirements of input variables and output product. Likewise, some types of HFUs listed in Table 1 may require different and complex computation methods to perform routing, in order to determine the runoff movement in a basin. For example, in a plain area, depending on flow regimes and characteristics, the plain river HFU can be simulated using a one-dimensional river model or a two-dimensional river model, as described in the third in this series of papers. For the lake and reservoir HFUs, such as in flood detention and retention ponds, they can be set as a zero-dimensional lake model if the only concern is water storage; when focusing on the flow characteristics in different zones, such as the velocity, they can be set as a two-dimensional lake model. The lake and reservoir HFU in a plain area also includes flood plains and paddy fields (Table 1), since water can flow into/out of these areas as storage units with relatively low velocities during a flood.

In the second of this series of papers (Distributed-Framework Basin Modeling System: II. Hydrologic Modeling System (II)), hilly sub-watershed and plain overland flow HFUs are described in detail, since they are the most common and frequently used in most regions. The plain river HFU, the lake and reservoir HFU, and the hydraulic structure HFU are introduced in the third in this series of papers (Distributed-Framework Basin Modeling System: III. Hydraulic Modeling System (III)), which focuses on the hydraulic calculation method in runoff concentration on underlying surfaces and flow movement in river networks and lakes. For urban pipe network HFU, unsaturated/saturated soil water zones HFU will be introduced in future papers.

The previous hydrologic unit or hydrological response unit is used as a discrete element for the simulation of a river basin [36], which is employed to describe the spatial variability for the topography, underlying surface, meteorological factor, and so on. In this study, we propose the concept of hydrologic computing units (HCUs) for the simulation of runoff generation and movement. The concept of HCU is similar to the previous hydrological response unit used in the Soil and Water Assessment Tool (SWAT) model [8], but HCUs are not only employed for basin discretization in the DF-HMS. In this study, the main representations of HCUs include subwatershed, sloped planes for overland flow, river cross-sections, and grids. For example, river cross-sections are HCUs for hilly river HFUs

or plain river HFUs in 1-D simulations. Computational grids are HCUs for plain overland-flow plane HFUs or saturated groundwater zone HFUs in 2D simulations. Moreover, each sub-watershed is considered to be an HCU for the hilly watershed HFU; in essence, each HFU can be further divided into small parts/zones when necessary.

For DF-PMS, HFUs can be further divided into HCUs in order to consider the spatial/characteristic changes. However, the discrete scale of the HFUs should be suitable for reflecting the temporal and spatial distribution of the hydrological cycle. Normally, the smaller the discrete scale of the HFU is, the higher the accuracy will be. For HFUs of the runoff generation type, the time scale usually ranges from an hour to a day, such as for the snowfield HFU, hilly sub-watershed HFU, and plain overland-flow HFU (Table 1). For HFUs of the routing type, the time scales range from seconds to hours, such as for the plain river HFU, the urban pipe network HFU, and the lake and reservoir HFU. The spatial scale depends on the time scale. Similarly, the spatial scale is a kilometer or larger for runoff generation HFUs. The spatial scale for HFUs of the movement type is much smaller, ranging from meters to kilometers. Overall, the optimal simulation discrete scale depends on the combination of different HFUs.

### 2.3. Systems Integration for GIS and the Professional Model System

Normally, GIS and professional model systems are integrated in a uniform interface, but coupling is based on the data exchanged. For convenience, the professional model is specified as the hydrological model below. Different kinds of hydrological models are mainly used to simulate dynamic water cycles that different from geographic information. However, all of the models in the corresponding traditional GIS are nonprocedural static models that show a deficiency with respect to the in-process dynamic model, which is the issue of temporal GIS. In this study, we propose a new way to solve the integration of GIS within the hydrological model. The double-object sharing structure (DOSS) is proposed in this study, which is designed through the self-developed GIS and professional models based on the underlying source code.

From a programming point of view, the hydrological model object and the GIS object partially have the same characteristics with respect to spatial, object, and basic data structure. The hydrological model object and GIS exhibit a certain overlap (Figure 4a), which is the theoretical basis for the combination of hydrological models and GIS. Therefore, the hydrological model and GIS can form one complex shared GIS object. The complex shared GIS object has the composite attribute of spatial features and the geographical process, which consists of a series of basic GIS objects. To solve the double-object information interaction between the hydrological model and GIS, the sharing GIS object needs to be built. Based on the existing structure, DOSS with the same geographic space is proposed (Figure 4b). The DOSS is able to interact with a particular category of data because it contains the original objects of the hydrological model and GIS. Due to the different requirements, it can be reorganized through different GIS and hydrological model objects to achieve a corresponding relationship. However, the data structures of the hydrological model and GIS are in an inconsistent state, so the sharing uniform data structure needs to be redefined to accomplish integration. There is a corresponding relationship between the object clusters in DOSS. We will take the two-dimensional river DOSS as an example, below.

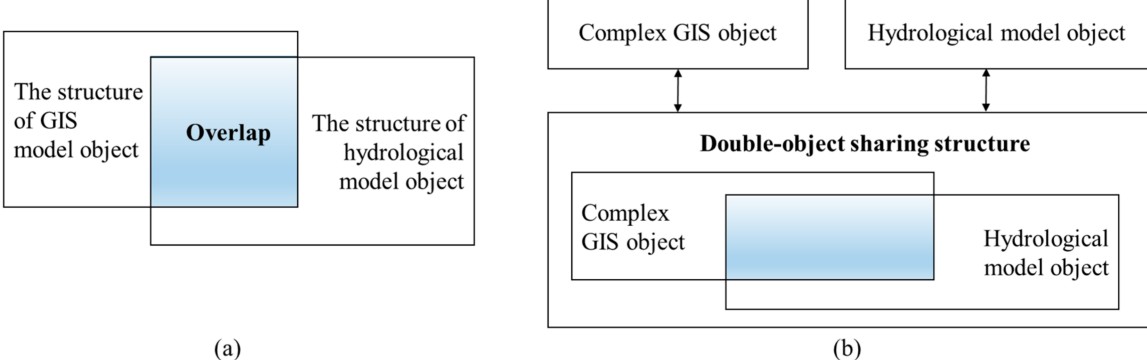

**Figure 4.** (**a**) The structure overlap of a GIS model object and a hydrological model object. (**b**) A diagram of the double-object sharing structure.

The complex DOSS generally has three parts: (a) The commonly needed data structure; (b) the necessary basic data structure for the representation of the GIS object; and (c) the data structure for the representation of professional model simulation results. The corresponding GIS object of the two-dimensional river model needs to reflect curvilinear grid information and query simulation results. In general, the simulations of the river contain the water level, depth, flow velocity, discharge, and other parameters. For the two-dimensional river model of a GIS object, it needs to represent the zone boundary of the river, the line of vertical flow direction, the line of parallel flow direction, terrain, the color and width of objective elements, etc. In the computation of the hydrological model, it needs the boundary line, the grid number, the node elevation, the node water depth, the node velocity (U, V), and to calculate the parameters derived from the basic grid. Hence, the hydrological model and GIS both need the boundary line; for example, the connected boundary information is multiplied in public classification (Table 2). The line of the parallel flow direction (such as X-coordinate, Y-coordinate) and bottom elevation (elevation of grid nodes) also belong to public classification. The velocity V, velocity U, water surface elevation Z, and concentration field are the private model objects. Similarly, the type, width, and pattern of line elements are the private GIS objects.

The data structure of DOSS can be clearly shown in the programming of a two-dimensional river. The public objects can be set as sharing arrays, such as m_LineNumber, M_NumofXgrid, M_NumofYgrid, m_XX, m_YY, and m_ZD. The two-dimensional river DOSS can easily express topological relations for the grid information, node water depth, node velocity (U, V), and nodes. The professional model and GIS objects display one-to-one correspondence inside, so the integration of GIS and hydrological models can be easily achieved. Even when the model is running, the DOSS can obtain and demonstrate information for both geographical and hydrological data at the same time. Through the method of DOSS implemented in DFBMS, the issue of temporal GIS is basically solved. DFBMS does not require data exchange when the model is running, which greatly improves the model efficiency.

**Table 2.** Parameters of the two-dimensional river model.

| Type of Variable | Name of Variable | Variable Information | Classify |
|---|---|---|---|
| CArray < UINT, UINT> | m_LineNumber | Multiply connected boundary information | Public |
| LONG | M_NumofXgrid | The total grid number in X direction | Public |
| LONG | M_NumofYgrid | The total grid number in Y direction | Public |
| Double | m_XX | X-coordinate | Public |
| Double | m_YY | Y-coordinate | Public |
| Double | m_ZD | Elevation of grid nodes | Public |
| Double | m_AVV | Velocity V | Model object |
| Double | m_AUU | Velocity U | Model object |
| Double | m_TZZ | Water surface elevation Z | Model object |
| Double | m_CTT | Concentration field | Model object |
| BOOL | m_bLineType | Type of line element | GIS object |
| UINT | m_LineWidth | Width of line element | GIS object |
| SymbolInfo | m_LSymbolInfo | The pattern of line element | GIS object |

## 2.4. Development of the Distributed-Framework Basin Modeling System

DFBMS has three parts: A database management system, a professional model system, and a geographical integrated system. The issue of coupling between the professional model system and GIS is solved by developing a DOSS. For coupling with the database management system, it is transmitted using generic database components. In terms of the system structure, the information flow is shown in Figure 5. The generic database interface component is like a bridge, connecting the database management system with the professional model system and double-object sharing structure GIS for the reading and writing of data. It can access many kinds of databases, such as ODBC, SQL Server, and ORACLE.

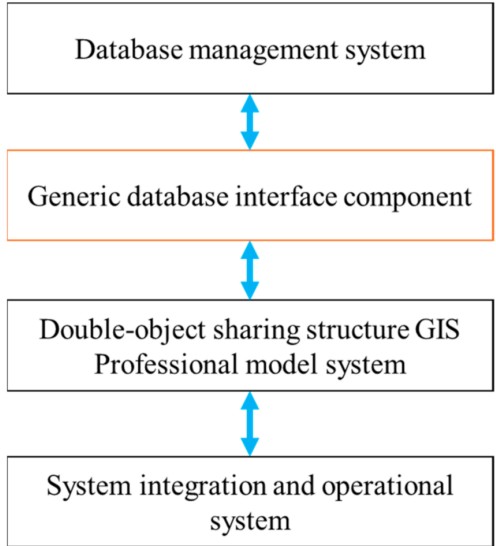

**Figure 5.** The information flow of DFBMS.

Based on the above study, DFBMS is programmed in Microsoft Visual C++. Visual modeling, program customization, dynamic queries, online analytic processing, and dynamic displays are involved in DFBMS. Based on the self-developed GIS, DFBMS integrates a professional model base, database, and GIS through the underlying sources code. It supports different format data sources, such as AutoCAD, ArcGIS, Mapinfo, user-defined, and so on. The model interface is shown in Figure 6. DFBMS also includes the professional model systems of water quality, water quantity, and sediment, but these are not mentioned in this series of papers. DFBMS can be used to simulate flow movement on the watershed scale or partial hydraulic engineering in a small region. The water movement and water

quality variety in DFBMS can be visualized in figures, tables, and animations. It can meet the practical requirements of flood-control planning, flood control impact assessment, real-time flood forecasting, water resource information management, water environmental assessment and protection, and so on.

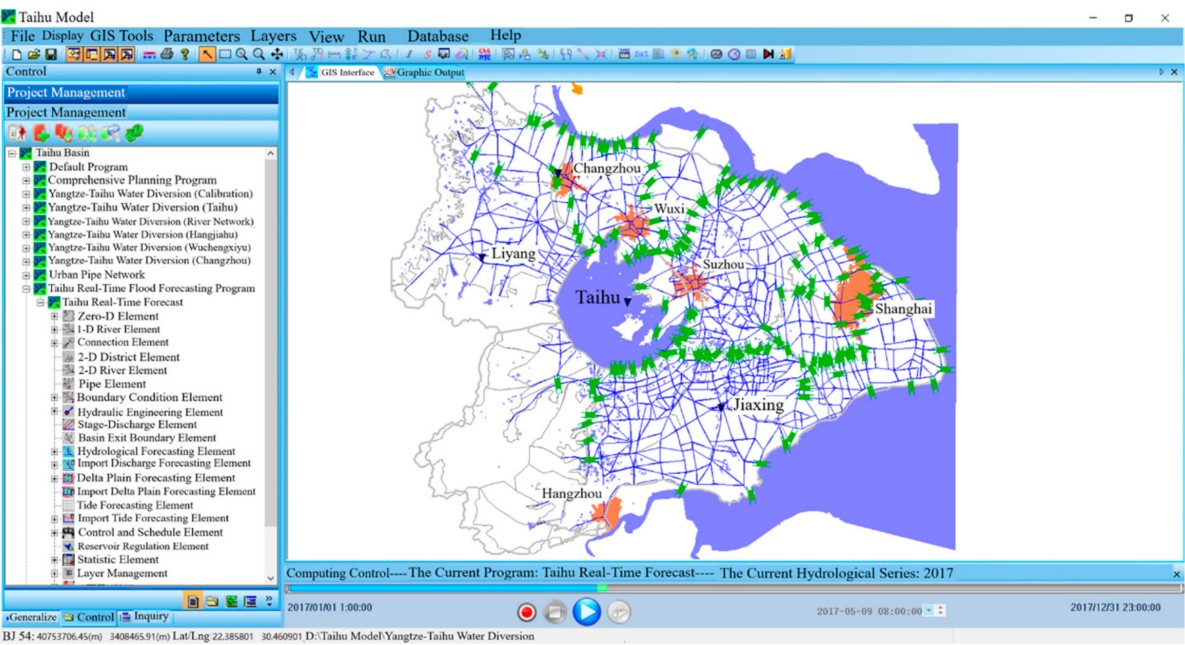

**Figure 6.** The interface of the distributed-framework basin modeling system.

## 3. Summary and Conclusions

DFBMS is an integrated hydrological and hydrodynamic model built to be used to simulate changes in the geographical environment in the past, present, and future. The structure of DFBMS consists of a professional model system, a double-object sharing structure GIS, and a database management system.

A distributed-framework professional modeling system (DF-PMS) is proposed that adopts different HFUs to simulate the whole watershed hydrological cycle. The HFU concept is the most important component of DF-PMS, enabling the model to simulate the hydrological process based on a conceptual formula or physical-based submodule. HFU is defined as a geographic area that has the same mechanism of runoff and confluence. The HFU can be classified into various runoff types, confluence types, or mixed types of runoff and confluence. With respect to the concept of the HFU, the distributed-frame professional modeling system in this study has 11 kinds of HFUs.

Overall, DF-PMS has the following characteristics: (1) The model has the capabilities of commonly used models based on the available data, including conceptual and distributed models, as well as black box, conceptual, physical-based, and topographic-based models; (2) the model has different numerical schemes for solving different problems; (3) the model can simulate different discretized spatial-scale watersheds; (4) the model can simulate most of the horizontal and vertical hydrology cycle of the basin; and (5) the model is easy to couple with the input of other models through a database management system, such as atmospheric migration models and land–air coupled models.

An double-object sharing structure (DOSS) is proposed to solve the deficiencies of GIS in-process dynamic models. In the DOSS, the sharing of a uniform data structure needs to be redefined to accomplish integration between the hydrological model and GIS. There is a corresponding relationship between the object clusters in DOSS. In the structure of double-structure GIS, new concepts of the complex geographic structure and expanding object structure are proposed to integrate applications in other fields, which is also the basis of DFBMS.

Based on the self-developed GIS, DFBMS integrates a professional model base, database management, and GIS through the underlying source code. DFBMS includes modules for water quality, water quantity, and sediment. The details of the hydrologic and hydraulic modeling system are described in the second and third paper in this series. Finally, the fourth paper verifies the feasibility and practicability of the DFBMS model on the basis of application in Taihu Basin.

**Author Contributions:** The work was conducted by C.W., W.H., G.C., X.F. and, X.L.; this paper was written by C.W.; G.C. and X.F. reviewed and improved the manuscript with comments; the data compilation and statistical analyses were completed by all authors. All authors have read and agreed to the published version of the manuscript.

**Funding:** This research has been financially supported by the National Key Research and Development Program of China (2018YFC1508200), Project 41901020 supported by NSFC, and the Fundamental Research Funds for the Central Universities (B200202030), and Hydraulic Science and Technology Program of Jiangsu Province (2020003).

**Institutional Review Board Statement:** Not applicable.

**Informed Consent Statement:** Not applicable.

**Data Availability Statement:** Not applicable.

**Conflicts of Interest:** The authors declare no conflict of interest.

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
