# Peer review of "Distributed-Framework Basin Modeling System: I. Overview and Model Coupling"

_water, doi:10.3390/w13050678_

Round 1
Reviewer 1 Report
The authors of the manuscript “Distributed-Framework Basin Modeling System: Overview and Model Coupling (Ⅰ)” made major alteration of their initial manuscript and have revised it according to the reviewers’ comments. The new output is an ameliorated version of the first manuscript. I recommend the minor revision of the manuscript based on the following comments:
General comments:
- The new revised version of the manuscript should be submitted including the track changes to better understand the changes. It is very difficult to understand where the alterations and amendments within the manuscript are located.
- The responses on the authors should clearly mention the lines of the alterations, otherwise the paper could be misjudged. This means that for each response the exact lines within the revised manuscript should be provided.
Specific comments:
Line 55: “model [9] and SWAT model [10].” At the end of this sentence the following sentence is proposed to be added: “Similarly, other spatial distributed models that simulate the hydrological cycle through dedicated inherent modules and have been applied in various spatial and temporal scales are also met in the literature (Habert et al., 2008; Skoulikaris et al., 2020)
- Habets, F., Boone, A., Champeaux, J. L., Etchevers, P., Franchisteguy, L., Leblois, E., ... & Viennot, P. (2008). The SAFRAN‐ISBA‐MODCOU hydrometeorological model applied over France. Journal of Geophysical Research: Atmospheres, 113(D6).
- Skoulikaris, C.; Anagnostopoulou, C.; Lazoglou, G. Hydrological Modeling Response to Climate Model Spatial Analysis of a South Eastern Europe International Basin. Climate 2020, 8, 1. https://doi.org/10.3390/cli8010001
Line 141: “The model of geological statistical analysis is….”. What the authors mean by geological? I suppose they want to say topographic or terrain? Please rephase.
In Lines 146-147 the authors state: “This represents the issue of temporal GIS—it is hard to extend the analysis function for GIS without solving the application of procedural dynamic models.” While in Lines 156- 158 the authors state “The double-structure GIS is proposed to solve the sharing issues between the complex geographical object and professional model object, which is a new way of exploring temporal GIS.” These two sentences are in controversy. Since the aim of the paper is to promote the temporal GIS function of their models, the authors cannot say that “…it is hard to extend the analysis function for GIS… “. Moreover in page 4 it is written “Through the method of DOSS, the issue of temporal GIS is solved.” however the specific argument is not supported within the manuscript. More information about the temporal GIS function should be added in the manuscript.
Lines 188-189: “Geographical Information System (GIS) is underlying developed which is the main part…”. What the authors mean with the expression “underlying developed”? There is no meaning, thus please rephase.
Line 38: The reference [5] is wrongly presented in the list of references. Please correct the reference properly.
Author Response
Thanks for your comments and suggestions. We have carefully considered all comments and made necessary revisions. Please check attached Word file.

Reviewer 2 Report
I recommended "accept" for it last time.
Author Response
Thank you for your support.
Reviewer 3 Report
I truly believe that the 4 papers could be synthesized into just one single paper, but I’ll leave that decision up to the Editor. Having said this, the authors have addressed each of my concerns and they are applauded for improving the papers. I appreciate the contribution of the authors in reporting their hydrologic modeling framework and I encourage them to work in future implementations with public repositories with codes and tutorials written in English for easier access to the international community.
Author Response
Thank you for your comments. We appreciate your review and help with our paper. We have discussed with the editor and get the permission from the editor to publish these four papers after revision. We are also working on the code publication and tutorials written in English. The DFBMS has been developed for more than decades, it is a sophisticated modeling system (framework) so that it will take some time and effort to complete above tasks.
This manuscript is a resubmission of an earlier submission. The following is a list of the peer review reports and author responses from that submission.
Round 1
Reviewer 1 Report
General comments about the series of papers
The authors of the series of manuscripts “Distributed-Framework Basin Modeling System: Overview and Model Coupling” present a new modelling approach. The so-called DFBMS model consists of a professional model system, a double-object sharing structure GIS, and a database management system. Although the significant work that the authors have conducted there are serious flaws that need to be addressed. In particular:
- The 1st paper presents an overall description of the model, however issues such as the functionality of the DOSS and the database management system are just presented without being analyzed. Thus I would recommend to focus only on the issues that the presented in the manuscripts 2, 3 and 4, i.e. the distributed-frame professional modeling system (DF-PMS), since DOSS and the database cannot be evaluated. I would recommend that these components to be limited to a couple of paragraphs.
- The main component of this new modelling approach is the hydrological feature unit (HFU). 11 different HFUs are presented, however the models that are interconnected within each HFU are not properly presented. There are some custom made models, however as it is understood some HFUs integrate model some existing models. This should be explicitly presented in a form of a table and for each HFU to explain the connected simulation models. Moreover, the table should have a column giving information where to find (in which section) the models description within the series of manuscripts.
- The GIS component should be better feet within the manuscript without generalities. As understood, the GIS component is being used for the delineation of a basins’ DEM, thus this should be the clear meaning of the GIS module demonstration.
- The authors claim that their approach is a spatiotemporal approach particularly with the use of the proposed GIS, but this is not proved. More information should be provided.
- The Complex geographic object section presents information that is not related or connected with the rest of the manuscripts.
- It’s hard to follow the manuscript flow, not in term of English language, but in terms of structure. For example, some HFUs are presented in the one manuscript, some others to another manuscript, without having a complete view of the processes. Thus I would recommend the reconstruction of the first manuscript.
- Issues related with potentialities, the use of Remote sensing or virtual reality or GPS are issues without any added value, since the just refereed and they are being presented.
- The advantages or shortcomings of the proposed models used in HFUs should be clearly stated in a form of a table. The authors try to propose a new approach without explicitly provide the benefits.
- Issues about the validation of the models should be more clear in the 2nd and 4th The 2nd manuscript need a lot of improvement.
- The GIS component claims that uses temporal components as an advantage. However this is not demonstrated at all.
Based on the aforementioned comments, as well as the comments that are attributed to each deliverable I recommend the major revision of the series of manuscripts.
General comments on the 1st manuscript.
Within the manuscript “Distributed-Framework Basin Modeling System: Overview and Model Coupling (Ⅰ)” the authors present a (very) general description of their proposed model. The model, namely DFBMS, consists of a professional model system, a double-object sharing structure GIS, and a database management system. The professional model system is responsible for the simulation of the various natural processes with the use of the so called “HFUs”. The double-object sharing structure, namely DOSS, is the interface for coupling GIS with the proposed simulation models, while the database management system is the database system where all the information are stored.
However, the principal aim and the added value of this model, as I understood, is the integration of the time component within the simulation of space processes. However, this issue is not well supported by the authors. A significant shortcoming is that the authors do not present a comparison table of their model’s potentiality against the capabilities of the existing models. This would be very important to support their arguments for producing a new model.
Various issues are presented without going into further details, such as the models that are used within HFUs, such as which are these models, did the authors developed the models or they integrate already known solutions, etc. The DOSS system description is based on discussion about objects integration, however this part of the document is lacking of references.
Overall, it’s obvious that the authors have put a lot of effort on creating the proposed model. I do also understand that this manuscript cannot be reviewed on its own, since is related with the other three submitted manuscripts. At any case, this first manuscript requires major revisions in order to be able to be a stand-alone manuscript.
Specific Comments 1.
The issue of “temporal GIS” i.e. the insertion of time in spatial data is an issue that could be also added in the abstract as a potential advantage of the newly proposed model.
A figure demonstrating the way that DFBMS, DF-PMS, DOSS, UFU are interconnected and form the logical structure of the proposed model should be added within the manuscript.
A list of abbreviations is rather required in the beginning of the manuscript, e.g.
DFBMS: distributed-framework basin modeling system
DF-PMS: distributed-frame professional modeling system
DOSS: double-object sharing structure.
HFU: hydrological feature unit.
etc
The authors use repeatedly the term “The distributed hydrological model…” while they want to speak more general about the functionalities of this type of models. Thus, the authors should revise the document to change the previous expression with “A distributed hydrological model…”, or with “Distributed hydrological models…”
Specific comments 2
Line 25. “DFBMS has many advantages compared with the previous hydrological model.” Which is the previous hydrological model? Please be more precise.
Line 32. “The watershed hydrological model…”. This sentence should be reworded as “A watershed hydrological model…” or “Watershed hydrological modelling….”
Line 33-34. “…proposed [1], the distributed hydrological model has been developed…”. This sentence it should be reworded as “…proposed [1], distributed hydrological models have been developed…”
Lines 36-37. “In the last few years, with the development of computer science, remote sensing, and GIS, the cost..”. This sentence should be reworded as “In the last few years, with the development of Information Communication Technologies (ICTs) (Skoulikaris et al. 2018), such as computing science, remote sensing and GIS, the cost..”
Proposed reference: Skoulikaris, Ch., Filali-Meknassi, Y., Aureli, A., Amani, A., Jiménez-Cisneros, B.E. (2018) Information-Communication Technologies as an Integrated Water Resources Management (IWRM) tool for sustainable development. In: Komatina, D. (eds) Integrated River Basin Management for Sustainable Development of Regions, InTech Publications.
Line 40. “and so on”. It should be reworded as “and other physical processes”.
Line 43. The word “normally” is not required.
Line 48. “…and so on”. The authors should reword this expression
Line 48. “Meanwhile, the distributed model describes” it should be reworded as “Meanwhile, a distributed model describes..”
Lines 49-51. “It is also easy to calibrate and can easily meet the requirement of the forecast, such as the distributed Xinanjiang model [8] and SWAT model [9].” This sentence is not connected with the previous sentences of the paragraph. Moreover, the meaning is not clear. Please reword and make a connection with the previous part of the paragraph.
Line 52: “Compared to the lumped model, the distributed model has obvious advantages..”. Which are these advantages? Authors should clearly state the advantages of the distributed models.
Lines 53-60. “In the Distributed Model Intercomparison Project (DMIP), … than the distributed model [10, 14].” The paper tries to introduce a new distributed hydrological model, thus the outputs of the specific project, since it is refereed in the manuscript, should be presented with more details in order to better understand the shortcomings of distributed modelling. What were the identified reasons within that project?
Line 96. What is HEC-SAS?
Lines 118-120. “In terms of traditional topological relations of GIS, it is very complicated to describe complex objects of hydrology and hard to satisfy the complex hydrological analyses.” Please provide examples or references for the aforementioned statement.
Line 122: “The model of geological statistical analysis is well-developed in GIS, which contains a digital elevation model,…” In which way the geological statistical analysis is related with the digital elevation model?
Line 143. “One is the information highway and high-speed wideband network technology…”. Is the model going to use on-the-fly data and produce real time simulations? For example, RS images are going to be utilized just after being accessed through a solid network connection? Please provide some more explanation on the use of the aforementioned statement.
Line 162. “Geographical Integrated system”. This part of the model has a different name than the one given in the Figure 1, namely 3S Integrated system. Which is the correct one?
Line 163. “It mainly contains the…” What do you mean that it contains RS or GIS or…? Does it contains modules responsible for GIS and RS analysis? Please be more precise.
Lines 169-171. “For GPS, information collection is the key point. In the 3S integrated system, it works through geographic information navigation.” It is not clear what the authors want to say. Please reword and be more precise of the use of GPS.
Line 171-172. “If there is no VR, visual representation cannot be achieved;” Are you going to use VR tools for the animation of the outputs? Which outputs? Or something else? Please be more precise about the use of VR within the model.
Line 175. “containing various professional models”. Which are these “professional” models? What do you exactly mean?
In Figure 1 the authors should state within the Professional model system: hydrological model and hydraulic model and optional use of the other models, since they are not being presented and thus evaluated within the specific research.
Lines 179-180: “DFBMS is not only designed for theoretical research, but also for solving practical problems. For example, the real-time requirement is an important feature in flood forecasting”. In which way the existing hydraulic models proceed with real time forecasting? Why not using a batch file for inserting prognosis data within an existing model and simulate next few days’ potential floods? What is the advantage of your model and in which way the aforementioned statement provides advantages?
Lines 232-235. “In this study, there are hilly sub-watershed HFUs, plain overland-flow HFUs, and unsaturated soil-water zone HFUs. There are hilly river HFUs, plain river HFUs, and lake and reservoir HFUs in the model due to the fact that the water movement in rivers, lakes, reservoirs, and flood retention areas is also different.” It cannot be understood which HFUs are proposed in the manuscript, the 3 first or the 4 others, or all together 7 HFUs? Please rephrase.
Lines 238-239. “The river and flood retention area are treated as two different HFUs, which are river HFUs and lake and reservoir HFUs”. What is the river HFUs? It is not referred within the table 1.
Lines 225-243. Which is the role of Snowfield HFU? Please add relevant information.
Lines 246-248. “one area can be simulated with the lumped model [29], and the other area can be simulated with the distributed Xinanjiang model [4].” Which is that lumped model?, add its name and description. Moreover, for common HFUs, e.g. hilly sub-watershed HFU, how many simulation alternatives the model offers? The 2 aforementioned? Are all the alternatives already developed models? This is a point that it should be clarified within the manuscript. And what about in the other HFUs? A table with the simulation alternatives for each HFUs should be added within the manuscript.
Line 248. “The simulation methods are chosen according to the requirements”. Which are these requirements? More information are needed.
Lines 251-252. “the plain river HFU can be simulated as a one-dimensional-river model or two-dimensional-river model.” These models are developed and proposed by the authors, or they are already (well-known) developed models?
Line 261. Since HCUs are used in the 2nd manuscript, more detailed information should be provided about them.
Lines 264-265. “In this study, the main representations of HCUs include subwatershed, sloped planes for overland flow, river cross-sections, and grids”. What are the grids?
Below Figure 3. Lines XXXX. “The major concern for the coupling of HFUs is the water exchange between the interface of different HFUs. When the water exchange occurs in one direction during the period or the variation of the time scale is large, the explicit coupling mode is applied for some interfaces of HFUs, such as exchange between the HFUs of rainfall, evapotranspiration, and infiltration, in the vertical simulation.” The aforementioned arguments are not clear. The sentences should be reworded and answers to the following questions should be provided: What are the interface(s) of different HFUs? Which is the exchange period? What is the “explicit coupling mode”? Which are the HFUs of rainfall, evapotranspiration, and infiltration? In which way the vertical simulation is used and what is the vertical simulation, since in this study as it is stated in Line 193, only the precipitation is used.
Lines XXX. “Normally, GIS and professional model systems are run independently, and are connected through data exchange [26].” The specific argument is rather outdated, since there are nowadays numerous models, where the GIS component is within the model system e.g. the new version of HEC-HMS.
Lines XXX. “For convenience, the professional model is specified as the hydrological model below. For different kinds of hydrological models, they are mainly dynamic models.” The meaning is not clear.
Line XXXX. “Based on the above study”. Which study do you mean? Please be more precise.
Conclusions. Line XXX. “DFBMS is a visual decision-making system that can be used to simulate change of the geographical environment in the past, present, and future.” DFBMS is presented in the manuscript as a distributed basin model which integrates GIS characteristics. Thus, it’s functionality as decision making system does not convert the model to a decision system. Please reword the 1st sentence of the conclusions in order to coincide with the rest of the manuscript.
Line XXX. “(4) the model can simulate the horizontal and vertical hydrology cycle of the whole basin”. This is in controversy with the statement that is given by the authors that the final product of the upper vertical layers, i.e. precipitation, is used and the other natural processes for the formation of precipitation are not taken into account. Reword this sentence.
Reviewer 2 Report
The authors try to describe a new structure of hydrological platform tool to simulate hydrological cycle (in water volume and water quality) taking in charge all the hydrological processes in relationship with changing complex geographic objects along the time, the DFBMS. The DFBMS platform consists on a DF-PMS (the model), a GIS system and a classic Data Management System. The innovation is in the DF-PMS based on the HFU concept (Hydrological Feature Concept) and the double-object sharing structure (the DOSS) to boost the classic GIS properties.
For this last innovation, the interest of the DOSS is well described, but I am not able to judge the validity of the demonstration provided.
For the first innovation, the HFU is the key principle of this model. First the acronym is not well chosen since it is very confused with the Hydrological Functional Unit commonly used from 20 years by the hydrologist. Second, the HFU concept is not well defined and the paper is not well organized to clearly demonstrate the interest and the functionality of these 11 possible HFUs.
For example, the authors wrote: The HFU is defined as a geographic area that has the same mechanism of runoff and confluence. What does it mean ?
Another point to underline is that the authors used a lot of assertion, such as: The model has the ability of commonly used models, including conceptual and distributed models, as well as black box, conceptual, physical-based, and topographic-based models. So, how it works ? No word to explain ?
The authors will find a lot remarks, comments in their manuscript. I hope they help them to define better their target and the pathway to achieve it. Sometimes, I was afraid that it could have some mistaken translation between Chinese and English: for example for "feature", "professional", "flow" and "runoff", "confluence" and "limit" or boundary",... It's not clear also to know if this model platform is for research demand (as written in page 13) of only for decision-making system as written in the conclusion.
Take care, the introduction 1.1 is not really a review. It's more a discussion on hydrological modeling issues. The authors would win to write a real introduction to precise their purpose. What do they want to demonstrate ?
They said at the beginning of the introduction that a model is a simplified representation of real world system. YES, but they may precise that the best model is dedicated to one purpose: the purpose could be the water resource management, the understanding of water processes, and so at different spatial and temporal scales. Then the models are often very specific. I understood the authors try to develop a generic model to be applied in a large range of situations, such as SHE, SWAT, MIKE-11... Then for some aims, we don't need to get more precise spatial scale, isn't it ? And what about the time scale they didn't mention ??? etc...
To conclude, I feel the paper is not enough precise and well organized, mainly about the role and need of HFU and HCU. How to justify these two objects ? Why no plain sub-watershed ? Why only plain overland flow ? etc.
At last, I guess you will get advantages to describe and to draw the algorithm used, i mean the logical framework of the hydrological pattern. And the bibliographic references are not enough recent. The youngest ref is from 2015, and 50% of the only 31 references were published more than 20 years ago. Then the bibliography should absolutely be refreshed.
Done to help.
The best

Reviewer 3 Report
The subject is current and very important. To better simulate the river basin hydrological cycle and solve practical engineering application issue, more and more precise techniques for simulating and understanding watershed hydrologic processes are being sought. The goal of this research was develop the theory and method of the distributed-framework basin modeling system (DFBMS). My basic remarks to the paper:
• An interesting proposal of a new approach to the analyzed problem was presented.
• Literature review is correct and contains basic items. The work contains a very detailed discussion of previous studies. This is the basic advantage of this paper.
• From the point of view of methodology, the paper lacks some more detailed, technical information about the construction of the new system / modeling approach.
• The results of the study were well analyzed.
• In my opinion, the summary can be expanded a bit and refer to the results in more detail.
• Figures 3 and 5 should be improved, are too simplistic and trivial.
• Paper must be comprehensively assessed to the entire series of articles.
The submitted paper made a good impression on me. The manuscript is well structured and deserves publication after some minor revisions. I congratulate you on taking up an interesting and important topic.
Reviewer 4 Report
This is a very solid manuscript. The authors did a good job at describing how DFBMS works. And as the first of four papers in series, it does give readers a very good idea about the functionality and feasibility of DFBMS. I don't have major concerns. My only comment is: I don't think Figure 5 is relevant to this study, although it helps explain the concept of geographic object. It doesn't make sense to me to specifically include these two figures in a hydrology paper.
Reviewer 5 Report
This series of 4 papers presents the description of a distributed basin modeling system composed of several components. The strength of the model laid on the flexibility and number of processes that can be integrated and modeled across the hydrologic and hydraulic components. In general, the papers are well-written, and the methods included in the hydrologic and hydraulic components are well presented. Having said this, I do not consider proper to present this work as a series of 4 papers, the overall structure looks more suitable for a dissertation document or a chapter book, but the presented format does not fit with the overall goal of a scientific paper, in which should maximize the synthetizes of methods, results, and discussions without losing accuracy and transparency, which reminds me the popular said: “I didn't have time to write a paper, so I wrote a book.” I encourage the authors to reconsider to condense the work into one single paper in order to show their wonderful work. Below, I’m describing my major and minor comments for all 4 papers.
[Major Comments]
- The 4 papers should be presented as a single paper. “Paper 1” can be easily synthesized as the introduction section, “Paper 2” and “Paper 3” is the section method, and “Paper 4” would be the Case Study. I noted through the papers several redundancies that can be avoided in order to achieve the best synthetizes in the work. For instance, “Paper 2” and “Paper 3” show a Case study; however, that should be the main purpose of “Paper 4”. There are several sections in “Paper 2” and “Paper 3” that can be moved to a Supplemental Material section (See Minor Comments).
- As I mentioned before, the strength of the study is found in the hydrologic and hydraulic components. However, “Paper 4” decreases the overall impact of the presented model. The authors were limited to show that the proposed model was able to replicate the discharge, water depth in some gauges just for a short period of time (calibration 1 year, validation 1 year). In general, there is not an analysis of the spatial distribution of the model performance, and there is no understanding of how the different model components, either hydrologic or hydraulic, is improving the representation of the hydrologic processes. This paper shows a new hydrologic model framework, therefore, should be expected to find an extended analysis of the different capabilities of the models showing the improvement of the model with and without different components
- The authors did not provide the source code or repository of the model. This is essential for future implementations of the model in the hydrologic community. In the case that the model is not available to the public, the authors must provide further details about the configuration in computational times used to run the simulations.
Minor Comments on: “Distributed-Framework Basin Modeling System: Overview and Model Coupling (I)”
[Line 25] What advantages? This statement is ambiguous
[Line 33] What does FH69 stand for?
[Line 106] Be careful using the argument of “Temporal GIS”, this is a matter of perspective, somebody could argue that including time-series to represent rainfall fields is sufficient to be in the realm of “Temporal GIS”. Besides, geological models do not seem an appropriate example to show the inconvenience of temporal representations in the current hydrological models, note that the geological processes evolve in a dramatic larger time scale in comparison to hydrologic processes explored in most of the hydrologic models.
[Figure 5] The captions need to be improved
Minor Comments on: “Distributed-Framework Basin Modeling System: Hydrologic Modeling System (II)”
[Line 20] Only two HFUs? what about the other 9 HFUs? Is there any model documentation?
[Line 122] What do SFD and MFD stand for?
[Section 2.1 2.1.1, 2.1.2] This section could be omitted or summarized in one or two paragraphs. Most of this content may be considered as general knowledge in the hydrologic community (e.g. estimation of flow direction D8), so there is no need to be so explicit in its development. Another option is included as Supplemental Material.
[Equation 2] What water depth and Chezy coefficient are used in Eq 2? Are they varying over space? Or is it just the DEM elevation used as the water depth in this case? Or are assumed constant across all the DEM processing?
[Lines 261-268] The runoff generation process and the overland flow must be explained in this section! The authors are just limited to provide some references; this is one of the key elements in the description of any hydrologic model.
[Lines 270-278] Be specific with the hydrograph method. This statement is vague, what equations and approximations are the authors using for the hydrograph routing method?
[Section 2.2.3] So does the plain overland runoff generation considers land use, but the Hilly- subwatersheds do not? Section 2.2.3 is nicely documented, however, section 2.2.1 is poorly described.
[Line 320] What specific parameter range? Be specific.
[Line 323] How necessary is to include this complexity in modeling runoff on the paddy fields? Have the authors provided any evidence of the adequacy in including this process? This should be explored in high detail in “Paper 4”
[Line 361] Provide ranges for Hp, Hu, and Hd
[Section 2.3] What about the modeling in woodland land use?
[Section 2.2.4] Include a description of the overland runoff method used. Again, what about the woodland surface?
[Section 2.3] This is not necessary if it is mentioned in “Paper 3”
[3 Study Case] This should be part of “Paper 4”
[Section 3.2] The evolution of the model performance needs to be improved. Please consider using the Nash–Sutcliffe model efficiency coefficient since has been used as a standard in the hydrologic community
[Section 3.2] Be specific in how the calibration was performed. What method? And what parameters were calibrated in this case study?
[Section 3.2] What was the computational time?
Minor Comments on: “Distributed-Framework Basin Modeling System: Hydraulic Modeling System (III)”
[Section 2] Large part of this paper could be included as an Appendix or Supplementals Information
[General] The equation numbering is incorrect, please verify.
[Lines 42-59] If there is no further discussion about these aspects through the paper, then this section should be removed.
[Section 2.4] This should be part of the “Paper 4”
[Line 443-Line 444] Rewrite “1982 cases...” for “case for the year 1982”; same for 1991.
[Figure 10] It is a better option to use a color bar to show the velocity field
Minor Comments on: “Distributed-Framework Basin Modeling System: Application in Taihu Basin (IV)”
[Figure 11] Are there only 4 streamflow gauges? Show statistics RMSE, Nash–Sutcliffe model efficiency
[Section 3.2] Why is the validation period and calibration period so short? I assume that there should be longer streamflow records within the basin, however, the authors only used one year for calibration and one for validation which obscures the true overall model performance that could be achieved with larger hydrologic records.
[General] Show the drainage area associated with each streamflow station